# Screening of Indicators to Evaluate the Overwintering Growth of Leaf-Vegetable Sweet Potato Seedlings and Their Main Influential Factors

Xiao Xiao [1], Xiaoju Tu [1], Kunquan Zhong [1], An Zhang [2] and Zhenxie Yi [1,*]

[1] College of Agronomy, Hunan Agricultural University, Changsha 410128, China; xiaoxiao@hunan.edu.cn (X.X.); tuxiaoju@hunan.edu.cn (X.T.)

[2] Changsha Xinwannong Seed Industry Co., Ltd., Changsha 410129, China; zhangruxun@163.com

* Correspondence: yizhenxie@hunan.edu.cn

**Abstract:** Whether the stems and leaves of leaf-vegetable sweet potatoes can be listed ahead of schedule is related to the improvement in economic benefits for farmers, and the key to all of this is to implement the safe overwintering of potato seedlings under the premise of saving production costs. Only in this way can we truly seize the "market opportunity" and achieve the goals of cost saving and increasing economic benefit. In this study, the main leaf-vegetable sweet potato variety Fucai 18 was used as the material, and the L9(3⁴) orthogonal experiment was carried out in a simple solar greenhouse environment for two consecutive years from 2021 to 2022 and from 2022 to 2023, respectively. The effects of nine different combinations of factors on the above-ground and underground agronomic traits of overwintering sweet potato seedlings were studied under the conditions of four factors and three levels: planting density (a); different cutting seedlings (b); rooting agent concentration (c); and transplanting time (d). The methods of principal component analysis, membership function method, cluster analysis, grey correlation degree and stepwise regression analysis were used to evaluate the growth of overwintering seedlings, and try to screen out the key indicators that can be used to identify and evaluate the growth of overwintering sweet potato seedlings. Through range analysis, identify the optimal combination of four factors and three levels, and explore the main factors that have a significant impact on the key indicators for evaluating the growth of overwintering potato seedlings. The results indicate the following: (1) The use of simple sunlight greenhouse in Changsha area can achieve the safe overwintering of vegetable sweet potato seedlings. (2) Stem thickness, root length, and root diameter can be used as three key indicators for identifying and evaluating the growth potential of vegetable sweet potato overwintering seedlings. (3) Under four factors and three levels, the best combination was A3B3C1D1 (planting density of 250,000 plants/ha, stem tip core-plucking seedlings, rooting agent concentration of 50 mg/L, the first batch of transplanting time). (4) The transplanting time (D) is the main factor for the two key evaluation indicators of stem diameter and root diameter, while there is no significant difference in the three other factors. (5) Different cutting seedlings (B) are the main influencing factors for the key evaluation index of root length, while the other three factors have the following impact on root length: transplanting time (D) > rooting agent concentration (C) > planting density (A). The results of this study not only contribute to the construction of a safe overwintering cultivation technology system for vegetable sweet potato seedlings, but also provide a certain theoretical basis for the breeding of new cold-leaf-vegetable sweet potato varieties in the future.

**Keywords:** leaf-vegetable sweet potato; L9(3⁴) orthogonal experiment; main influencing factors; overwinter seedlings; valuating indicator

## 1. Introduction

The leaf-vegetable sweet potato (*Ipomoea batatas*) is a new variety of sweet potato in which the fresh and tender leaves, petioles, and tender stems approximately 10–15 cm

below the growth point of the stem tip are used as edible vegetables. In recent years, with the improvement in living standards and the transformation of views of consumption, leaf-vegetable sweet potatoes have gradually become the "new favorite" of farmer's markets in China and throughout the world [1,2]. Leaf-vegetable sweet potatoes grow rapidly and can be picked for the first time after 30 days of cutting. On average, it can be picked once every 7–10 days in a normal year, with a long picking period, high yield, and good benefits [3]. Experiments have shown that under sufficient water and fertilizer supply, continuous picking for 5–6 times can yield 34,500–45,000 kg of stems and leaves per hectare. Calculated at RMB 3 per kilo (the earlier the listing time, the higher the price), the profit per hectare after deducting costs is approximately RMB 30,000–60,000 [4,5]. Sweet potatoes prefer warmth and are sensitive to low temperatures. When the temperature decreases to <15 °C, the aboveground parts stop growing. When the temperature drops to 6–8 °C, the leaves begin to wilt and turn brown, and they quickly wither and die due to frost [6]. Affected by the low temperature in spring and cold spells in late spring, most regions of China (including Changsha) need to wait until the average daily temperature reaches 15 °C in mid-to-late April each year. At this time, sweet potatoes can only be planted in the open field. It takes about 30 days, that is, in mid-to-late May, for the stems and leaves of leaf-vegetable sweet potatoes to achieve large-scale market sales. However, at this time, all kinds of vegetables have been listed in large quantities and the supply is sufficient, resulting in a significant drop in market prices. For the majority of leaf-vegetable sweet potato farmers, it is obvious that they have lost "business opportunities" and failed to maximize economic benefits. It should be pointed out that the vegetable supply in most areas of China is in the "empty window period" from February to April every year. During this period, both the variety and supply of vegetables were unable to meet market demand, resulting in consistently high vegetable prices. Therefore, the vast number of vegetable sweet potato growers should seize the "market opportunity" in a timely manner, strive to achieve the early listing and sales of stems and leaves, in order to obtain better economic benefits. Compared with the traditional cultivation mode, the overwintering cultivation of sweet potato for vegetables can be put on the market 20–30 days earlier because it omits the link of seeding and breeding. In addition, the high-frequency harvesting of stems and leaves results in the majority of assimilation products and inorganic nutrients being transported to the aboveground parts of the stems and leaves, leading to low yield and poor quality of seed potatoes, which seriously restricts the cultivation, expansion, and sales of leaf-vegetable sweet potatoes in the second year. In summary, in order to achieve the annual production of leaf-vegetable sweet potatoes and advance the long-term and sufficient market supply of stems and leaves, it is necessary to carry out the winter cultivation of leaf-vegetable sweet potatoes. However, according to the local winter climate conditions, sweet potatoes in Changsha cannot safely overwinter in the wild and must rely on sunlight greenhouses. The sunlight greenhouses built by vegetable farmers in our country to save production costs are relatively simple, generally without heating equipment, and can only maintain the "life-saving" state of crops. The indoor temperature during the day is often below 20 °C, and at night it drops to 5–12 °C, which undoubtedly brings certain difficulties to the overwintering of vegetable sweet potato seedlings. This is consistent with the suboptimal temperature regime cultivation that has been promoted and advocated in many countries [7], which aims to achieve cost saving, energy reduction, efficiency improvement, and green production.

At present, many scholars focus their research on the breeding of new varieties, cultivation techniques, fertilizer application, and the taste quality of Leaf-vegetable sweet potatoes [8–12], while research on overwintering cultivation techniques is very limited, mainly focusing on cultivation density. The selection of cutting seedlings, cutting time, low-temperature exercise, and cold-resistant variety screening were studied in [13–17]. There have been no reports on the comprehensive evaluation of overwintering leaf-vegetable sweet potato seedling growth under multiple factors using multivariate statistical analysis methods, and the key indicator system for identifying and evaluating the overwintering

growth of leaf-vegetable sweet potatoes has not yet been constructed. This study utilized Fucaishu 18, the main variety of Leaf-vegetable sweet potato in various provinces in the middle and lower reaches of the Yangtze River, as the experimental material. In a solar greenhouse, an orthogonal study L9($3^4$) with four factors and three levels was conducted in Changsha, and it examined the planting density, the selection of cutting seedlings, different concentrations of rooting agents, and transplanting time. And, multiple statistical methods, such as correlation analysis, principal component analysis, membership function analysis, stepwise regression, and cluster analysis were used to comprehensively evaluate the agronomic traits of overwintering sweet potato seedlings for vegetable use. Key indicators that can be used for the identification and evaluation of overwintering leaf-vegetable sweet potato seedling growth were selected. At the same time, the range analysis is used to explore the best combination of factors and the primary and secondary effects of key indicators of overwintering seedling growth. We hope that this experiment can provide theoretical and technical support for the safe overwintering of leaf-vegetable sweet potato seedlings without increasing the production cost, and realize the early marketing of stems and leaves, as well as provide basic data for the further in-depth study of cold tolerance mechanism.

## 2. Materials and Methods

### 2.1. Experimental Materials

Fucaishu 18 (Quanshu 830 × Tainong 71), which was selected by the Crop Institute of the Fujian Academy of Agricultural Sciences (Fujian, China), is currently the primary variety of vegetable sweet potato cultivated in the middle and lower reaches of the Yangtze River.

### 2.2. Introduction to Experimental Location

Changsha City is located in the northern part of eastern Hunan Province in the lower reaches of the Xiangjiang River and the western edge of the Changliu Basin, with a longitude of 111°53′–114°15′ E and a latitude of 27°51′–28°41′ N. It has a subtropical monsoon climate with a winter that has a duration of approximately 117–122 days and an average temperature of 4.4–5.1 °C. Due to its terrain and the interaction between warm and cold air, the weather is inclement and less sunny in the winter. For example, from early January to middle February 2022, Changsha was the provincial capital with the least amount of sunshine in China, with a cumulative duration of sunshine of merely 24.2 h. The daily duration of illumination time < 1 h. On this basis, it can be concluded that vegetable sweet potato seedlings cannot be safely overwintered outside in Changsha, and the plants must be grown under insulated conditions indoors with the help of solar greenhouses. Given the weak economic strength of the vast number of farmers in our country, the construction of solar greenhouses to save production costs is relatively simple, and there is no heating equipment, which can only maintain the "life saving" state of crops. The indoor temperature is often lower than 20 °C during the day, and the indoor temperature drops to 5–12 °C at night [18].

### 2.3. Experimental Design

The experiment was conducted in the vegetable base of Xinwannong Seed Industry Co., Ltd., Hehu Formation, Chunhua Mountain Village, Chunhua Town, Changsha City, Hunan Province, from 1 November 2021 to 20 February 2022 and from 21 October 2022 to 28 February 2023, respectively. The greenhouse was a steel frame structure, and the film was made of polyethylene, 0.07 mm thick, and transmitted more than 90% of the light. The soil in the greenhouse had a moderate level of fertility, with 21.16 g/kg of organic matter, 0.93 g/kg of total nitrogen, 20.94 mg/kg of available phosphorus, 153.29 mg/kg of available potassium, and a pH of 5.57. The study adopted an orthogonal design L9($3^4$) with four factors and three levels (Table 1). The four factors included the planting density (A), transplanted seedlings (B), different concentrations of rooting agents (C), and transplanting time (D). There were nine treatments (Table 2) at the three levels, three replicates, and a

random block arrangement with a total of 27 plots. Each plot was 2.5 m long, 1 m wide, and covered an area of 2.5 m².

**Table 1.** The factors and levels of L9(3⁴) orthogonal experiment.

| Levels | Factor | | | |
|---|---|---|---|---|
| | Density (Ten Thousand Plants/ha) A | Transplanted Seedlings B | 98% Indolebutyric Acid C | Transplanting Time D |
| 1 | 16 (0.3 m × 0.27 m) | The seedlings of stem tip | 50 mg/L | First batch |
| 2 | 20 (0.3 m × 0.23 m) | Midstem seedling | 75 mg/L | Second batch |
| 3 | 25 (0.3 m × 0.18 m) | Shoot tip coring Seedling | 100 mg/L | Third batch |

**Table 2.** L9(3⁴) orthogonal design [19].

| Treatment Group | Factor | | | | |
|---|---|---|---|---|---|
| | A | B | C | D | Combination |
| 1 | 1 | 1 | 1 | 1 | A1B1C1D1 |
| 2 | 1 | 2 | 2 | 2 | A1B2C2D2 |
| 3 | 1 | 3 | 3 | 3 | A1B3C3D3 |
| 4 | 2 | 1 | 2 | 3 | A2B1C2D3 |
| 5 | 2 | 2 | 3 | 1 | A2B2C3D1 |
| 6 | 2 | 3 | 1 | 2 | A2B3C1D2 |
| 7 | 3 | 1 | 3 | 2 | A3B1C3D2 |
| 8 | 3 | 2 | 1 | 3 | A3B2C1D3 |
| 9 | 3 | 3 | 2 | 1 | A3B3C2D1 |

Healthy and pest-free sweet potato seedlings from the field were cut to approximately 20–25 cm. Before cutting, the bases of sweet potato seedlings were soaked in 98% indolebutyric acid (IBA) for 60 min. For stem tip seedlings, the front part of the sweet potato vine had six internode lengths, and the growth point of the stem tip was not removed. Thus, the top leaf was retained. Stem tip pick seedling was cut the same as described above with 6 internode lengths, while the growth point of the stem tip was removed. The top leaf was retained. For the middle-stage seedlings, the upper middle part of the sweet potato vine was cut and selected with six internodes and no leaves. It was important to ensure that the three nodes were inserted into the soil. Based on the preparation of land, 4500 kg/ha of commercial organic fertilizer (organic matter content ≥ 30%) and 700 kg/ha (17-5-26) of commercial compound fertilizer were applied uniformly. After 30 days from the time that the sweet potato seedlings were cut, 0.5% urea plus 0.1% potassium dihydrogen phosphate foliar fertilizer was sprayed continuously on the leaves three times every 7 days. During the overwintering period, when the daily average outdoor temperature < 10 °C, the door was tightly closed to maintain and increase the indoor temperature. An automatic temperature and humidity recorder (RC-4, Jiangsu Jingchuang Electric Co., Ltd., Jiangsu, China) was hung in the center of the greenhouse, and the temperature inside the greenhouse was recorded every 60 min.

*2.4. Measurement Indicators and Methods*

2.4.1. Root Traits

The root traits were analyzed using an EPSON Expression 11000XL 3.49 root scanner (Epson [China] Co., Ltd., Beijing, China) and the WinRHIZO 2012 Root System Professional Edition (Regent Instruments, Quebec City, QC, Canada).

2.4.2. Rate of Survival

The calculation for the rate of survival is as follows:

$$(\text{Number of surviving seedlings/Cutting seedlings}) \times 100\% \tag{1}$$

### 2.4.3. Root Activity

The root activity was measured with triphenyltetrazolium chloride (TTC) as previously described and shown below [20]:

$$\text{Tetrazolium (TTC) reduction intensity (mg)}/[\text{g (fresh root weight)}/\text{h}] = \text{TTC reduction amount (mg)}/[\text{root weight (g)} \times \text{Time (h)}] \tag{2}$$

### 2.4.4. Root–Shoot Ratio

The root–shoot ratio was determined as follows:

$$\text{Fresh weight of the belowground roots/Fresh weight of the aboveground stems and leaves} \tag{3}$$

### 2.4.5. Stem Diameter

A Vernier caliper was used to measure the stem diameter of the upper, middle, and lower parts of the main stems of the sweet potato seedlings, and the mean values were taken.

### 2.4.6. Number of Leaves

All the unfolded functional leaves above the ground were counted.

### 2.5. Statistical Analysis

Microsoft Excel 2016 was used to organize and analyze the data, and SPSS 26.0 (IBM, Inc., Armonk, NY, USA) was used for variance analysis, principal component analysis, membership function analysis, stepwise regression and cluster analysis. Relevant indicators are calculated as follows [21]:

(1) The Principal Components from the PCA Were Extracted Based on the Criteria That the Eigenvalue > 1 or the Sum of Principal Components > 80%

(2) The Membership Function Value Is Shown as Follows:

$$u\,(Xj) = (Xj - Xmin)/(Xmax - Xmin);\ i = 1, 2, 3, \ldots, n \tag{4}$$

where *Xj* represents the *j*th comprehensive indicator, while *Xmin* and *Xmax* represent the minimum and maximum scores of each trait indicator on each principal component, respectively.

(3) The Weight Was Calculated as Follows:

$$wj = pj / \sum_{j}^{n} pj;\ wj \text{ represents the weight of the } j\text{th principal component} \tag{5}$$

where *pj* represents the eigenvalue corresponding to the extracted principal component.

(4) The Comprehensive Evaluation Value of Growth Was Calculated as Follows:

$$D = \sum_{j}^{n} u[(Xi) \times Wi] \tag{6}$$

where *D* represents the comprehensive evaluation value of the growth of overwintering sweet potato seedlings in each treatment group. Clustering statistics were formed on the *D* value and classified the overwintering growth of sweet potato seedlings at different levels of combinations of factors based on the *D* value.

## 3. Results

### 3.1. Temperature Changes Inside the Greenhouse during Overwintering

In the 2-year winter experiment of vegetable sweet potato seedlings, the trend of changes in the indoor temperature in the solar greenhouse was basically consistent during the 2-year overwintering study. There were four stages of changes, including "rapid

cooling—brief warming—cooling again—gradual warming." As shown in Figure 1, from early November of that year, as the weather gradually cooled and the temperature continued to decrease, the indoor temperature also began to gradually decrease. A cliff-like decrease occurred in late November. By the middle of December, the indoor temperature decreased to approximately 10 °C. From late December to early January of the following year, although the indoor temperature briefly increased with the rise of external temperature, it quickly decreased again. The period from mid-to-late January to early February is the coldest period of the year in the Changsha area, and it is also the period when the temperature inside the greenhouse is the lowest. Afterwards, the indoor temperature gradually rebounded with the increase in the external temperature.

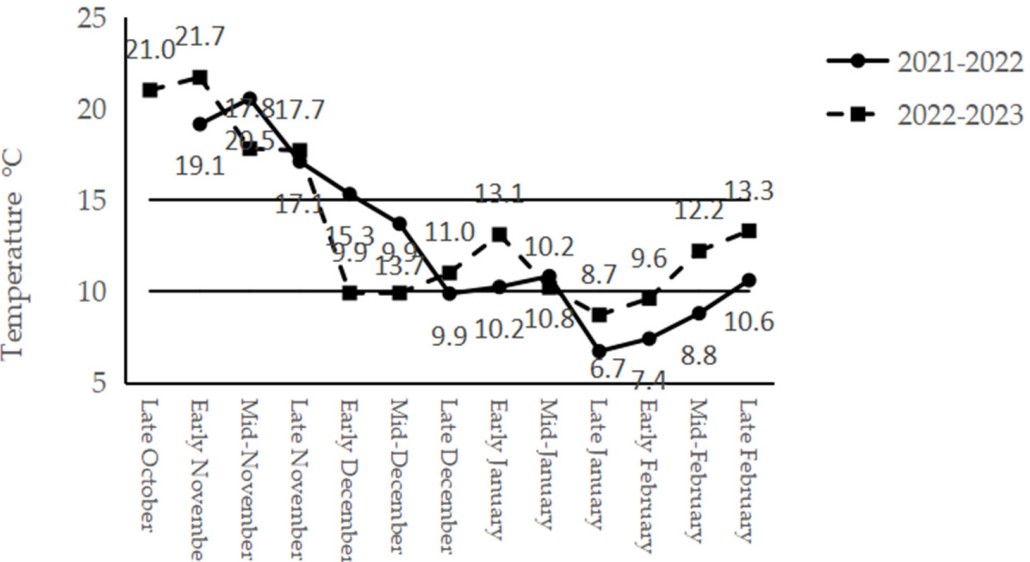

**Figure 1.** Temperature changes inside the greenhouse during overwintering.

According to Table 3, during the coldest period inside the greenhouse, which occurred between middle January and early February each year, there were 13 out of 31 days when the temperature inside the greenhouse once reached or exceeded 15 °C in 2022, with an average duration of 5.0 h. The longest duration of 7.5 h occurred on 5 February 2022, and the shortest duration of only 1 h occurred on 3 February 2022. In 2023, there were 15 days when the temperature inside the greenhouse once reached or exceeded 15 °C, with the longest duration of 12 h (13 January 2023) and the shortest duration of 4 h (17 January 2023). The average duration was 6.9 h, and it was primarily concentrated between 10:00 and 17:00. During this period, the highest temperatures in the greenhouse reached 33.7 °C (at 14:00 on 19 January 2022) and 33 °C (at 14:00 on 13 January 2023), while the lowest temperatures in the greenhouse were as low as −0.3 °C (at 7:00 on 5 February 2022) and −3.5 °C (at 7:00 on 25 January 2023). There were nine consecutive days in 2022 (21 January to 29 February 2022) when the temperature inside the greenhouse did not reach or exceed 15 °C, and 4.9 °C ≤ daily average temperature ≤ 9.5 °C. Similarly, during this period in 2023, there were 9 days (2 February to 10 February 2023) when the temperature inside the greenhouse did not reach or exceed 15 °C, and 3.5 °C ≤ daily average temperature ≤ 11.4 °C. As shown in Table 3, the difference in diurnal temperature in the greenhouse increased during this period, by an average of 11.5 °C and 14.5 °C in both years, respectively. The maximum diurnal temperature difference in 2022 occurred on 5 February (30.4 °C), while the maximum diurnal temperature difference in 2023 occurred on 28 January (34.4 °C).

In summary, although the vegetable sweet potato seedlings suffered from some degree of sustained low temperature stress during the overwintering period, they did not always remain below the critical temperature for sweet potato growth (15 °C) due to the insulating effect of the solar greenhouse.

**Table 3.** Daily average temperature, temperature difference, and duration above 15 °C in greenhouse (11 January–10 February 2022, 11 January–10 February 2023).

| | 2022 | | | 2023 | | |
|---|---|---|---|---|---|---|
| **Date** | **Average Temperature in Shed** | **Difference in Temperature** | **Duration ≥ 15 °C (h)** | **Average Temperature in Shed** | **Difference in Temperature** | **Duration ≥ 15 °C (h)** |
| | **°C** | **°C** | | **°C** | **°C** | |
| 11 January | 11.9 | 27.6 | 5.0 | 13.8 | 15.9 | 8.0 |
| 12 January | 12.1 | 26.9 | 7.0 | 17.5 | 16.8 | 11.0 |
| 13 January | 12.8 | 26.5 | 7.0 | 18.4 | 23.6 | 12.0 |
| 14 January | 8.6 | 4.2 | | 9.4 | 7.4 | 7.0 |
| 15 January | 10.1 | 9.8 | 4.0 | 3.5 | 3.9 | |
| 16 January | 8.9 | 4.4 | | 4.6 | 11.7 | |
| 17 January | 10.1 | 9.4 | 1.5 | 7.4 | 24.6 | 4.0 |
| 18 January | 12.5 | 28.2 | 6.5 | 8.3 | 26.4 | 6.0 |
| 19 January | 11.5 | 30.2 | 6.0 | 10.7 | 26.7 | 6.0 |
| 20 January | 9.8 | 14.3 | 3.0 | 8.6 | 18.6 | 7.0 |
| 21 January | 9.5 | 5.7 | | 9.2 | 6.4 | |
| 22 January | 7.4 | 6.4 | | 7.9 | 3.2 | |
| 23 January | 6.0 | 3.1 | | 7.7 | 2.8 | |
| 24 January | 5.9 | 3.3 | | 7.7 | 21.5 | 5.0 |
| 25 January | 6.4 | 4.1 | | 8.4 | 27.3 | 6.0 |
| 26 January | 6.6 | 3.8 | | 8.5 | 7.7 | |
| 27 January | 6.1 | 3.5 | | 8.4 | 20.2 | 6.0 |
| 28 January | 4.8 | 2.9 | | 8.0 | 34.4 | 7.0 |
| 29 January | 4.7 | 7.4 | | 8.9 | 34.1 | 8.0 |
| 30 January | 7.8 | 16.4 | 3.5 | 9.4 | 26.8 | 0.0 |
| 31 January | 6.1 | 9.2 | | 12.1 | 28.6 | 9.0 |
| 1 February | 7.3 | 9.2 | | 12.8 | 12.6 | 8.0 |
| 2 February | 5.0 | 2.6 | | 9.3 | 6.0 | |
| 3 February | 7.2 | 13.8 | 1 | 6.6 | 4.2 | |
| 4 February | 10.0 | 3.6 | 6 | 7.4 | 7.2 | |
| 5 February | 11.4 | 30.4 | 7.5 | 8.4 | 4.3 | |
| 6 February | 8.4 | 6.6 | | 9.6 | 5.9 | |
| 7 February | 5.6 | 7.2 | | 10.4 | 4.0 | |
| 8 February | 5.4 | 6.0 | | 11.4 | 7.1 | |
| 9 February | 8.0 | 21.4 | 5.5 | 10.4 | 4.2 | |
| 10 February | 7.4 | 8.6 | | 9.2 | 5.1 | |
| Average | 8.2 | 11.5 | 5.0 | 9.5 | 14.5 | 6.9 |

*3.2. Effects and Correlation Analysis of the Different Treatments on the Primary Agronomic Traits of the Overwintering Seedlings*

In this study, the states of growth of the overwintering seedlings of the vegetable sweet potatoes treated with nine different factor combinations were distinguished based on to their rate of survival and eight major agronomic traits, including their root length, root area, and root activity, among others (Table 4). The results showed that Treatment Group 1 (A1B1C1D1) had the highest rates of survival and four agronomic traits, including root length, root area, root volume, and leaf number. These values were all significantly higher than those of the treatment group that ranked second ($p < 0.05$). Treatment Group 9

(A3B3C2D1) ranked first in terms of root diameter and stem diameter, and there was a significant difference ($p < 0.05$) compared to the treatment group that ranked second. Treatment Group 3 (A1B3C3D3) ranked first in terms of the root–shoot ratio, and there were significant differences between the treatment groups ($p < 0.05$). The highest value of root activity was observed in Treatment Group 7, but there was no significant difference between the Treatment Groups 1, 6, 8, and 9 ($p > 0.05$). All three replicates of the overwintering seedlings in Treatment Group 2 (A1B2C2D2) died during the overwintering period, and their related indicators were designated 0 and included in the subsequent statistical analysis.

**Table 4.** Effects of different factors on the growth characteristics of overwintering seedlings of vegetable sweet potatoes.

| Processing Number | Survival Rate (%) | Root Length (mm) | Root Area (cm²) | Root Diameter (mm) | Root Volume (cm³) | Root Vitality (μg/min/g) | Root–Canopy Ratio | Thick Stem (mm) | Number of Blades |
|---|---|---|---|---|---|---|---|---|---|
| Treatment 1 | 45.000 ± 4.082 a | 268.043 ± 10.752 a | 100.647 ± 3.546 a | 1.192 ± 0.057 b | 3.048 ± 0.183 a | 4.945 ± 0.739 a | 0.237 ± 0.0001 h | 6.823 ± 1.732 ab | 50.667 ± 5.033 a |
| Treatment 2 | 0 g | 0 h | 0 h | 0 e | 0 e | 0 d | 0 i | 0 d | 0 f |
| Treatment 3 | 17.500 ± 2.143 c | 196.844 ± 7.874 d | 39.430 ± 1.389 f | 0.658 ± 0.033 d | 0.646 ± 0.039 d | 2.469 ± 0.071 c | 1.541 ± 0.0145 a | 6.820 ± 0.550 bc | 7.333 ± 1.528 e |
| Treatment 4 | 22.917 ± 2.807 b | 254.892 ± 10.196 b | 50.781 ± 1.523 d | 0.637 ± 0.032 d | 0.806 ± 0.048 d | 3.432 ± 0.341 bc | 1.055 ± 0.0001 b | 5.710 ± 0.500 bc | 13.000 ± 2.646 de |
| Treatment 5 | 5.557 ± 1.961 f | 165.739 ± 6.629 f | 79.670 ± 2.390 b | 1.184 ± 0.059 b | 1.826 ± 0.110 c | 3.258 ± 0.325 c | 0.307 ± 0.0004 e | 6.920 ± 0.600 ab | 10.000 ± 3.000 de |
| Treatment 6 | 13.197 ± 2.557 cde | 105.104 ± 4.204 g | 32.742 ± 0.982 g | 0.752 ± 0.038 c | 0.644 ± 0.055 d | 4.666 ± 0.483 a | 0.386 ± 0.0007 d | 5.003 ± 0.435 c | 19.333 ± 3.512 c |
| Treatment 7 | 8.333 ± 1.359 ef | 212.441 ± 8.498 c | 45.709 ± 1.371 e | 0.707 ± 0.035 cd | 0.803 ± 0.048 d | 5.438 ± 0.167 a | 0.279 ± 0.0002 f | 6.090 ± 0.530 bc | 36.000 ± 3.055 b |
| Treatment 8 | 11.667 ± 1.400 de | 176.930 ± 7.077 ef | 39.681 ± 1.190 f | 0.706 ± 0.035 cd | 0.714 ± 0.043 d | 4.334 ± 0.556 ab | 0.588 ± 0.0002 c | 6.400 ± 0.560 bc | 14.667 ± 3.512 d |
| Treatment 9 | 15.553 ± 1.100 cd | 183.316 ± 7.333 e | 72.941 ± 2.188 c | 1.270 ± 0.064 a | 2.410 ± 0.145 b | 4.499 ± 1.008 a | 0.264 ± 0.0001 g | 8.070 ± 0.700 a | 34.000 ± 3.606 b |
| Average value | 15.529 | 173.785 | 51.289 | 0.790 | 1.211 | 3.671 | 0.558 | 5.699 | 21.593 |
| Standard deviation | 12.649 | 78.246 | 28.446 | 0.378 | 0.956 | 1.657 | 0.465 | 2.299 | 15.497 |
| variation coefficients | 0.815 | 0.450 | 0.555 | 0.478 | 0.789 | 0.451 | 0.833 | 0.523 | 0.714 |
| F value | 63.872 ** | 342.636 ** | 746.955 ** | 242.309 ** | 343.584 ** | 28.201 ** | 29,826.345 ** | 27.538 ** | 75.222 ** |

"**" indicated an extremely significant difference ($p < 0.01$) at 0.01 levels, respectively. Different letters in the same column indicate significant differences between different treatments ($p < 0.05$); different letters in the same column indicate significant difference ($p < 0.05$).

An additional analysis showed that the variation coefficient of each indicator was significant under each treatment. Among them, the variation coefficient of six indicators, including the survival rate, root area, root volume, root–shoot ratio, stem diameter, and leaf number, >50%, which indicated strong variation. This indicated that different treatments have significant impacts on the overwintering seedlings of the leaf-vegetable sweet potatoes, with large differences in the state of growth.

A correlation analysis was then conducted to accurately understand the states of growth of the overwintering seedlings of the vegetable sweet potatoes at different levels of treatment under various factors and the interrelationships between the nine indicators. The results indicated that there were clear correlations between the various indicators, but there were clear differences in their significance, correlation strength, and positive–negative relationships. As shown in Table 5, the number of leaves is positively correlated with the number of roots and the root diameter ($r > 0.75$, $p < 0.01$). There were highly significant positive correlations between seven groups of indicators, including the number of roots and root diameter, stem diameter, survival rate; root diameter and root length, stem diameter; root length and stem diameter; stem diameter and rate of survival ($0.3 < r < 0.75$, $p < 0.01$). There were significant positive correlations between five groups of indicators, including the number of roots and root length, root diameter and rate of survival, root length and the number of leaves, stem diameter and the number of leaves, the number of leaves, and rate of survival ($0.3 < r < 0.75$, $p < 0.05$). There was a weak positive correlation between the rate of survival and root length ($r < 0.3$, $p > 0.05$).

As shown in Table 6, there were highly significant positive correlations between five groups of indicators, including the root area and root volume, stem diameter; root diameter and root volume, stem diameter; root activity and the number of leaves ($r > 0.75$, $p < 0.01$). There were highly significant positive correlations between four groups of indicators, including the stem diameter and root length, root area, root activity, and the number of leaves and root volume ($r > 0.75$, $p < 0.05$). There were significant positive correlations between eight groups of indicators, including the survival rate and root length, root area, root volume; root length and root area, root activity; root area and the number of leaves; root diameter and root activity and the number of leaves ($0.3 < r < 0.75$, $p < 0.05$). There

were positive correlations between 12 groups of indicators, including the survival rate and root diameter, root activity, stem diameter, the number of leaves; root length and root diameter, root volume, root activity, root–shoot ratio, the number of leaves; root area and root activity; root volume and root activity, stem diameter; stem diameter and the number of leaves ($0.3 < r < 0.75$, $p > 0.05$). There were weak positive correlations between the root–shoot ratio and survival rate and the stem diameter ($r < 0.3$, $p > 0.05$). There were weak negative correlations between the root–shoot ratio and four indicators, including the root area, root diameter, root volume, and root activity ($-0.3 < r < 0$, $p > 0.05$). The root–shoot ratio negatively correlated with the number of leaves ($-0.75 < r < -0.3$, $p > 0.05$).

**Table 5.** Correlation analysis between indicators under different processing (2021–2022).

| | Number of Roots | Root Diameter | Root Length | Thick Stem | Number of Blades | Survival Rate |
|---|---|---|---|---|---|---|
| Number of roots | 1 | 0.729 ** | 0.493 * | 0.598 ** | 0.903 ** | 0.642 ** |
| Root diameter | | 1 | 0.717 ** | 0.618 ** | 0.760 ** | 0.541 * |
| Root length | | | 1 | 0.362 ** | 0.591 * | 0.144 |
| Thick stem | | | | 1 | 0.597 * | 0.663 ** |
| Number of blades | | | | | 1 | 0.587 * |
| Survival rate | | | | | | 1 |

"*" and "**" indicated significant differences ($p < 0.05$) and extremely significant differences ($p < 0.01$) at 0.05 and 0.01 levels, respectively.

**Table 6.** Correlation analysis between indicators under different processing (2022–2023).

| | Survival Rate | Root Length | Root Area | Root Diameter | Root Volume | Root Vitality | Root-Canopy Ratio | Thick Stem | Number of Blades |
|---|---|---|---|---|---|---|---|---|---|
| Survival rate | 1 | 0.726 * | 0.676 * | 0.489 | 0.671 * | 0.447 | 0.192 | 0.449 | 0.664 |
| Root length | | 1 | 0.744 * | 0.631 | 0.57 | 0.673 * | 0.413 | 0.796 * | 0.614 |
| Root area | | | 1 | 0.925 ** | 0.945 ** | 0.603 | −0.072 | 0.789 * | 0.732 * |
| Root diameter | | | | 1 | 0.883 ** | 0.695 * | −0.071 | 0.892 ** | 0.687 * |
| Root volume | | | | | 1 | 0.509 | −0.256 | 0.664 | 0.776 * |
| Root vitality | | | | | | 1 | −0.07 | 0.774 * | 0.804 ** |
| Root–canopy ratio | | | | | | | 1 | 0.272 | −0.325 |
| Thick stem | | | | | | | | 1 | 0.588 |
| Number of blades | | | | | | | | | 1 |

"*" and "**" indicated significant differences ($p < 0.05$) and extremely significant differences ($p < 0.01$) at 0.05 and 0.01 levels, respectively.

We found that the correlation coefficient between some indicators > 0.8 and was sometimes even > 0.9. This may be owing to the information overlap caused by multicollinearity. Thus, additional analysis and evaluation are merited.

### 3.3. Principal Component Analysis

The PCA was conducted on overwintering vegetable sweet potato seedlings in different treatments. Table 7 shows the relevant values of the principal component quantity, eigenvalues, rate of contribution, and cumulative rate of contribution during the 2021–2022 and 2022–2023 seasons.

**Table 7.** Principal component characteristic vector and contribution rate.

| | 2021–2022 | | 2022–2023 | | |
|---|---|---|---|---|---|
| **Principle Factor** | **1** | **2** | **1** | **2** | **3** |
| Eigen value | 4.031 | 0.969 | 5.614 | 1.493 | 0.920 |
| Contribution ratio (%) | 67.176 | 16.153 | 62.378 | 16.585 | 10.221 |
| Cumulative contribution ratio (%) | 67.176 | 83.329 | 62.378 | 78.963 | 89.184 |

### 3.4. Comprehensive Evaluation of the Growth States of Overwintering Vegetable Sweet Potato Seedlings in Each Treatment Group

3.4.1. Calculation of the Membership Function

As shown in Table 7, and according to formula (4), calculate the membership function values of each processing comprehensive indicator, and according to formula (5), calculate the weights of the principal component comprehensive indicator as 0.806, 0.194 (2021–2022) and 0669, 0.186, and 0.115 (2022–2023), respectively.

3.4.2. Comprehensive Evaluation Values (D) and Classification of the Overwintering Sweet Potato Seedlings in Each Treatment Group

The comprehensive evaluation values (D) of growth were calculated based on the weights and membership functions of the principal component comprehensive indicators and sorted by size. A larger D value indicated the better growth of the overwintering seedlings in the treatment group. As shown in Table 8, Treatment Groups 1 and 9 ranked first and second, respectively, in the studies during both years. Treatment Group 2 ranked the ninth, with the lowest comprehensive evaluation value (D) among the nine treatment groups, while Treatment Group 6 remained unchanged in its ranking. Although the rankings of the other five treatment groups had changed, they were relatively close to each other. For example, Treatment Groups 7 and 8 ranked sixth and seventh, respectively, in the study during the first year, and seventh and sixth, respectively, in the second year. Treatment Groups 3, 4, and 5 just changed their positions in the rankings 3–5.

**Table 8.** Comprehensive index value, weight, U (Xj), D value, and comprehensive evaluation of each treatment group.

| Treatment Group | 2021–2022 | | | | 2022–2023 | | | | |
|---|---|---|---|---|---|---|---|---|---|
| | U(X1) | U(X2) | D Value | Ranking | U(X1) | U(X2) | U(X3) | D Value | Ranking |
| Treatment 1 | 0.893 | 0.662 | 0.840 | 1 | 0.920 | 0.135 | 0.106 | 0.653 | 1 |
| Treatment 2 | 0.071 | 0.959 | 0.243 | 9 | 0.000 | 0.000 | 0.276 | 0.032 | 9 |
| Treatment 3 | 0.429 | 0.075 | 0.360 | 5 | 0.455 | 0.983 | 0.431 | 0.537 | 4 |
| Treatment 4 | 0.516 | 0.335 | 0.481 | 3 | 0.519 | 0.790 | 0.309 | 0.529 | 5 |
| Treatment 5 | 0.46 | 0.559 | 0.479 | 4 | 0.613 | 0.221 | 0.790 | 0.542 | 3 |
| Treatment 6 | 0.226 | 0.346 | 0.249 | 8 | 0.439 | 0.284 | 0.725 | 0.430 | 8 |
| Treatment 7 | 0.342 | 0.323 | 0.338 | 6 | 0.535 | 0.277 | 0.614 | 0.480 | 7 |
| Treatment 8 | 0.234 | 0.353 | 0.259 | 7 | 0.486 | 0.507 | 0.748 | 0.506 | 6 |
| Treatment 9 | 0.531 | 0.536 | 0.532 | 2 | 0.729 | 0.133 | 0.600 | 0.582 | 2 |
| Weight | 0.806 | 0.194 | | | 0.669 | 0.186 | 0.115 | | |

### 3.5. A Gray Correlation Analysis between the Growth Traits and D Value of the Overwintering Seedlings

As shown in Table 9, the correlation between the five indicators and the D values from 2021 to 2022 was in the order of the root diameter > root length > the number of roots > the number of leaves > stem diameter. The correlation coefficient ranged from 0.634 to 0.818 with a distance of only 0.184. The correlation between the eight indicators and the D values from 2022 to 2023 was in the order of the stem diameter > root length > root diameter > root activity > root area > number of leaves > root volume > root–shoot ratio. The correlation coefficient ranged from 0.661 to 0.922, with a distance of only 0.261. The two indicators of root diameter and root length ranked in the top two in the studies during the two years, which can obviously reflect the states of growth of the overwintering vegetable sweet potato seedlings.

**Table 9.** Analysis of the gray correlation between each individual indicator and the D value.

| | Correlation Coefficient and Ranking | | | |
| | 2021–2022 | | 2022–2023 | |
|---|---|---|---|---|
| Number of roots | 0.714 | 3 | | |
| Root diameter | 0.818 | 1 | 0.831 | 3 |
| Root length | 0.723 | 2 | 0.861 | 2 |
| Stem diameter | 0.628 | 5 | 0.922 | 1 |
| Number of blades | 0.634 | 4 | 0.697 | 6 |
| Root area | | | 0.793 | 5 |
| Root volume | | | 0.675 | 7 |
| Root–canopy ratio | | | 0.661 | 8 |
| Root Vitality | | | 0.823 | 4 |

*3.6. Establishment of the Regression Models and Screening of the Indicators of Overwintering State of Growth*

The comprehensive evaluation value (D) of the growth state of the overwintering vegetable sweet potato seedlings during the two years was taken as the dependent variable, and the index coefficient was taken as the independent variable for the stepwise regression analysis. The optimal regression equations were obtained as follows:

$$\text{PV (2022–2023)} = 0.024 \text{ stem diameter} + 0.001 \text{ root length} + 0.017 \text{ root activity} + 0.085 \text{ root diameter} + 0.036 \ (R^2 = 0.991, \ p < 0.001, \ F = 284.026) \tag{7}$$

The Durbin–Watson statistic was d = 1.593.

$$\text{PV (2021–2022)} = 0.059 \text{ root diameter} + 0.004 \text{ number of leaves} + 0.018 \text{ root length} + 0.04 \text{ number of roots} - 0.40 \ (R^2 = 0.991, \ p < 0.001, \ F = 1023.904) \tag{8}$$

The Durbin–Watson statistic was d = 1.571.

After the multicollinearity had been removed, both the regression equations showed two indicators, including the root length and root diameter. These equations were then used to predict the comprehensive index of the state of growth of the overwintering seedlings in each treatment group and compare this index with the original data (Table 10). As a result, the highest accuracy was 99.84%, and the lowest also reached 88.89%. In combination with the results of the gray correlation analysis, the root length and root diameter can be taken as two important agronomic traits to evaluate the state of growth of the overwintering seedlings of the vegetable sweet potatoes. In the study, during 2022–2023, the gray correlation coefficient of the stem diameter ranked first among the eight indicators. Although this indicator only ranked fifth among the five indicators in the preliminary study (2021–2022), it was only 0.184 less than the correlation coefficient of root diameter that ranked first, which is not a large difference. Therefore, it should also be included in the evaluation indicator system. Moreover, the first two involved the belowground root index of the overwintering sweet potato seedlings. The addition of the aboveground part index of the stem diameter ensured that the entire evaluation system was more balanced and complete, which made it easier to observe and evaluate.

**Table 10.** Analysis of evaluation accuracy of equation.

| Processing Group | 2021–2022 | | | 2022–2023 | | |
| | Predicted Value | Primary Value | Evaluation Accuracy (%) | Predicted Value | Primary Value | Evaluation Accuracy (%) |
|---|---|---|---|---|---|---|
| A1B1C1D1 | 0.84 | 0.829 | 98.69 | 0.654 | 0.653 | 99.84 |
| A1B2C2D2 | 0.243 | 0.253 | 96.05 | 0.036 | 0.032 | 88.89 |
| A1B3C3D3 | 0.36 | 0.346 | 96.11 | 0.497 | 0.537 | 92.55 |
| A1B3C3D4 | 0.481 | 0.471 | 97.92 | 0.54 | 0.529 | 97.96 |
| A1B3C3D5 | 0.479 | 0.483 | 99.17 | 0.554 | 0.542 | 95.49 |
| A1B3C3D6 | 0.249 | 0.248 | 99.60 | 0.423 | 0.430 | 98.37 |
| A1B3C3D7 | 0.338 | 0.328 | 97.04 | 0.525 | 0.480 | 91.43 |
| A1B3C3D8 | 0.259 | 0.241 | 93.05 | 0.597 | 0.506 | 97.49 |
| A1B3C3D9 | 0.532 | 0.546 | 97.44 | 0.654 | 0.582 | 99.84 |

### 3.7. Cluster Analysis

To further verify whether the three selected agronomic traits can truly reflect the state of growth of the overwintering seedlings, the Euclidean distance (shortest distance method) was used to perform systematic clustering analysis and K-means clustering analysis on the data in the studies during both years using the three indicators.

As shown in Figure 2a, the nine treatment groups can be divided into five categories at a distance of 2.5. Class I included Treatment Group 1 (A1B1C1D1). Class II included Treatment Group 2 (A1B2C2D2). Class III included Treatment Group 3 (A1B3C3D3) and Treatment Group 7 (A3B1C3D2). Class IV included Treatment Group 6 (A2B3C1D2) and Treatment Group 8 (A3B2C1D3). Class V included Treatment Group 4 (A2B1C2D3), Treatment Group 5 (A2B2C3D1), and Treatment Group 9 (A3B3C2D1).

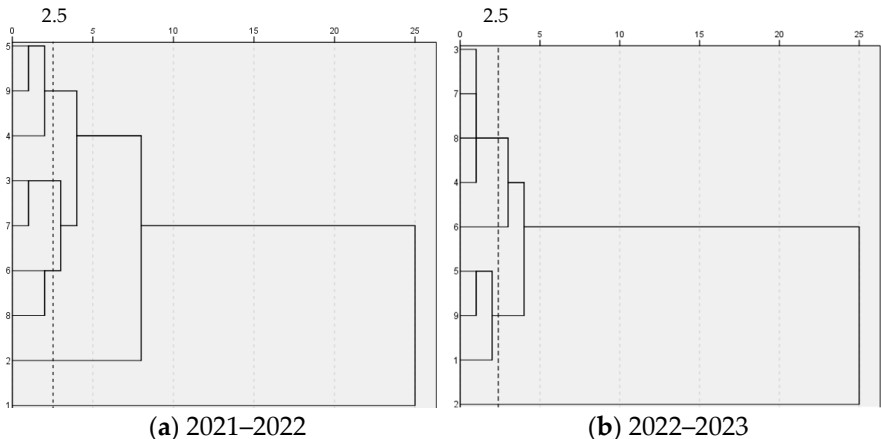

(**a**) 2021–2022 (**b**) 2022–2023

**Figure 2.** Fuzzy clustering dendrogram of the growth trend of overwintering sweet potato seedlings in 9 different leaf-vegetable treatments based on D-values.

The nine treatment groups in Figure 2b are divided into four categories. Class I included Treatment Group 1 (A1B1C1D1), Treatment Group 5 (A2B2C3D1), and Treatment Group 9 (A3B3C2D1). Class II included Treatment Group 2 (A1B2C2D2). Class III included Treatment Group 6 (A2B3C1D2). Class IV included Treatment Group 3 (A1B3C3D3), Treatment Group 4 (A2B1C2D3), Treatment Group 7 (A3B1C3D2), and Treatment Group 8 (A3B2C1D3). The classification of each treatment was completely consistent with the ranking of comprehensive evaluation D value. This analysis provides strong evidence that the three agronomic traits that were selected can serve as important indicators to identify the state of growth of the overwintering seedlings of the vegetable sweet potatoes.

### 3.8. Comprehensive Analysis of the Orthogonal Experiments

3.8.1. The Optimal Combination of the Three Evaluation Indicators for the State of Growth of the Overwintering Seedlings

As shown in Table 11, Factor A is at the third level in each of the three optimal combinations of indicators. Therefore, the optimal level of Factor A was A3. Factors B, C, and D all covered two levels (Level 1 and Level 3) in each of the three optimal combinations of indicators. The method of Lv [22] was used to compare and select the three remaining factors to determine the optimal level of each factor:

$$\text{Factor B: Stem diameter: (B1} - \text{B3)/B1} \times 100\% = -3.9\% \text{ (no advantage)} \tag{9}$$

$$\text{Root length: (B1} - \text{B3)/B1} \times 100\% = 34.1\% \text{ (having an advantage)} \tag{10}$$

$$\text{Root diameter: (B1} - \text{B3)/B1} \times 100\% = -5.75 \text{ (no advantage)} \tag{11}$$

It is apparent that B3 has a greater advantage over B1. Thus, the optimal level of Factor B is B3, which indicates that the stem tip topping seedlings should be selected for cutting.

$$\text{Factor C: Stem diameter: (C1 − C3)/C1} \times 100\% = -7.6\% \text{ (no advantage)} \tag{12}$$

$$\text{Root length: (C1 − C3)/C1} \times 100\% = 11.6\% \text{ (having an advantage)} \tag{13}$$

$$\text{Root diameter: (C1 − C3)/C1} \times 100\% = 3.7\% \text{ (having an advantage)} \tag{14}$$

It was apparent that C1 had a greater advantage over C3. Thus, the optimal level of Factor C was C1, which indicated that the optimal concentration of rooting agent was 50 mg/L.

$$\text{Factor D: Stem diameter: (D1 − D3)/D1} \times 100\% = 15.7\% \text{ (having an advantage)} \tag{15}$$

$$\text{Root length: (D1 − D3)/D1} \times 100\% = -1.8\% \text{ (no advantage)} \tag{16}$$

$$\text{Root diameter: (D1 − D3)/D1} \times 100\% = 45.1\% \text{ (having an advantage)} \tag{17}$$

It was apparent that D1 had a greater advantage over D3. Thus, the optimal level of Factor D was D1, which was the first batch of cuttings (21 October 2022)

In view of this analysis, A3B3C1D1 is the optimal combination for the orthogonal study of the overwintering seedlings of vegetable sweet potatoes.

**Table 11.** Statistical table of extremely poor analysis results of cold resistance indicators.

| Indicators | Optimal Combination | Effects of Various Factors (Duncan) |
|---|---|---|
| Thick stem | A3B3C3D1 | D(a) > A(b) > B(b) > C(b) |
| Root length | A3B1C1D3 | B(a) > D(b) > C(c) > A(d) |
| Root diameter | A3B3C1D1 | D(a) > A(b) > B(b) > C(b) |

3.8.2. Analysis of the Primary and Secondary Effects of the Four Factors

As shown in Table 11, the ranking of the primary and secondary effects of the four factors for the two indicators of the stem diameter and root diameter was completely consistent, and the primary effect Factor D (the batch of cutting) was significantly greater than that of the other three factors, and there was no significant difference between all the factors. There were significant differences among the four factors of root length, with the primary effect identified as Factor B (cutting location), followed by Factor D (the batch of cutting), Factor C (concentration of rooting agent), and Factor A (cutting density). It was apparent that Factor D (transplanting time) and Factor B (different cutting seedlings) were the primary effects of the three indicators, followed by Factor A (planting density) and Factor C (rooting agent concentration).

## 4. Discussion

### 4.1. Characteristics of Temperature Variation in Simple Solar Greenhouse during Winter

A simple solar greenhouse can be effective at increasing the temperature and keeping the plants warm, particularly on sunny days in the winter when the sun rises, and the temperature of the greenhouse increases rapidly. In such cases, the greenhouse reached its highest temperature at approximately 14:00 in the afternoon. The temperature and humidity records indicated that once the highest temperature reached 33 °C, it then decreased gradually as the sun set. In abnormal circumstances, it would decrease to <0 °C at approximately 7:00 the next day. This is consistent with the findings by Gao [23] that the temperature inside the greenhouse can go through four stages, including a rapid increase, slow increase, stable maintenance, and rapid decrease with the increase in the sunrise angle and light intensity on sunny days. In addition, as shown in Table 3, there was a large difference in the diurnal temperature in the greenhouse on sunny days in the winter, with a daily difference in the average diurnal temperature of 14.5 °C. This can help to improve the rate of the growth of plants in the greenhouse, promote root growth, and prevent foliar burns. Moreover, the temperature difference in the greenhouse is recommended to

be between 10 and 17 °C [24,25]. Therefore, using a simple greenhouse can enable the vegetable sweet potato seedlings in Changsha to overwinter safely. However, when the daily average temperature is below 10 °C or on clear winter nights, it is still necessary to take measures to prevent frigid temperatures and keep the plants warm. In addition to tightly closing the greenhouse, measures, such as covering the surroundings and the roof of the greenhouse with straw, and building warm soil walls can also be used to maintain a higher temperature.

*4.2. Selection of Evaluation Indicators for the Growth State of the Overwintering Seedlings of Vegetable Sweet Potatoes*

At present, researchers have conducted research on the response of various crops to low temperature stress in physical, chemical, morphological, yield, and other aspects, and proposed various indicators and methods for identifying plant cold tolerance based on this. Previous evaluations of the cold resistance of melon vegetables were mainly divided into two parts: the germination stage and the seedling stage. The germination stage was mainly evaluated based on indicators related to seed germination ability, while the seedling stage was mainly evaluated based on indicators related to plant damage, growth, and physiology [26]. The cold tolerance of plants during the seedling stage is usually evaluated in terms of two aspects: agronomic traits and physiological indicators. Due to the complexity of physiological and molecular level identification operations, appearance morphology is often used as an identification indicator for evaluating plant cold tolerance in the actual production processes. Morphological indicators can intuitively reflect the damage and tolerance of plants under low temperature stress. By studying the changes in plant phenotype after low temperature stress, the identification indicators related to cold tolerance can be screened. Du Wen li et al. [27] found that bitter gourd plants exhibit phenomena such as the wilting of true leaves, dehydration, and the significant softening and downward bending of petioles after low temperature stress. Zhou JianHui et al. [28] found that the fresh aboveground weight and dry root weight have a significant impact on the low temperature tolerance of sweet melons. The leaf area, aboveground fresh weight, and underground fresh weight of melon seedlings will significantly decrease with the prolongation of low temperature stress time [29]. The study also found that under low temperature and weak light conditions, the fresh root quality, root length, number of leaves, and root surface area of different melon varieties can be used as indicators for identifying their stress resistance [30]. Therefore, the above indicators can be selected as important indicators for identifying crop growth and cold tolerance in a low-temperature stress environment.

Plants do not passively respond to low temperature stress, but actively form a series of defense mechanisms at the tissue, physiological, cellular, and molecular levels through physiological and biochemical changes and responses, as well as the expression and regulation of related genes, ultimately manifested through a series of indicators such as morphology, physiological, and biochemical characteristics, and yield [31,32]. For many years, numerous scholars have used the grading method for stem and leaf damage under low temperature stress, as well as the determination of indicators such as antioxidant enzymes, osmoregulatory substances, and photosynthetic physiological characteristics. They have also used various evaluation methods such as phase relationship analysis, membership function analysis, and principal component analysis to screen a batch of sweet potato varieties with strong cold tolerance (including vegetable sweet potato varieties). Although they have also selected some physiological and biochemical indicators that can be used for cold tolerance evaluation, such as POD, APX, soluble sugars, and soluble proteins [13–18], their identification methods are too complex. With the deepening of research on the safe overwintering of vegetable sweet potatoes and the urgent need for production practice, it is necessary to screen out the relevant appearance indicators to make a more intuitive, accurate, and rapid evaluation of the growth of overwintering sweet potato seedlings.

This study selected multiple indicators of growth traits to evaluate the overwintering of vegetable sweet potato seedlings, and each indicator showed extremely significant or significant differences under different combinations of the four factors and three levels. The correlation analysis showed that most of the indicators also had varying correlations, which indicated that there was a collinear overlap between the information intersection and overlap between the indicators. Since plant stress resistance is a complex comprehensive trait, the genetic control of these traits has polygenic characteristics, and it is regulated by multiple gene loci [33]. Thus, single physiological or trait indicators cannot accurately and objectively evaluate the tolerance of crops to cold. Further research is merited using methods, such as a PCA, membership functions, clustering, and stepwise regression. These methods have been widely used in this study of the heat tolerance of major grain crops, such as corn (*Zea mays*), rice (*Oryza sativa*), and wheat (*Triticum aestivum*) [21,34,35]. A PCA was used to transform the original individual indicators into several independent and representative comprehensive indicators. A stepwise regression analysis established the optimal regression equation as follows:

$$PV\ (2022–2023) = 0.024\ \text{stem diameter} + 0.001\ \text{root length} + 0.017\ \text{root activity} + 0.085\ \text{root diameter} + 0.036 \tag{18}$$

$$PV\ (2021–2022) = 0.059\ \text{root diameter} + 0.004\ \text{number of leaves} + 0.018\ \text{root length} + 0.04\ \text{number of roots} - 0.40 \tag{19}$$

A comprehensive evaluation of the (D) value, predicted value (PV), and cluster analysis indicated that these findings were highly accurate. Three selected indicators cover important growth traits in the aboveground and belowground parts of the overwintering seedlings of vegetable sweet potatoes, which avoided the limitation and one-sided evaluation of cold tolerance that can occur when this tolerance is evaluated by single indicators. Compared with the physiological and biochemical methods of determining indicators, this method is simple and intuitive. The corresponding cultivation measures can be directly taken with regard to these three indicators in production, and the measures are practical.

*4.3. The Influences of Different Factors and Levels on the Indicators Used to Evaluate the State of Growth of the Overwintering Seedlings of Vegetable Sweet Potatoes*

Among the three indicators used to evaluate the state of growth of the overwintering seedlings of vegetable sweet potatoes selected in this study, two (root diameter and root length) originated from the belowground roots. This proves that the key to the safe overwintering of vegetable sweet potatoes is promoting the early establishment of root morphology and ensuring that they have a good state of growth. Moreover, healthy roots in a good state of growth provide a basis for thriving stems and leaves aboveground and a higher rate of survival of the plants. Research has shown that there are many factors that affect the rooting of cuttings, including biological characteristics, cutting location, hormones and other internal factors, as well as external factors, such as temperature, light, water, cutting period, and density [36]. According to the results of this study, the cutting materials and cutting period are two key factors. The diameter, length, and the number of leaves of plant cuttings have impacts on their rooting. As the cuttings become thicker, the cambium of young cells in the thickens, and the plant is able to survive much better. However, if the diameter of the cuttings is too large, the degree of lignification will increase, which does not facilitate rooting [37]. The research results of Wei indicated that the rooting rates of different branches at the top, middle, and base of *Daphne odora* var. *marginata* decreased from the top to bottom [38]. Studying sweet potatoes led some researchers to arrive at a similar conclusion that the stem tips of the overwintering sweet potato seedlings grew faster and better than their middle and base parts, and the aboveground stems are thicker than those of the potato seedlings with no stem tips [39,40]. However, this study came to a different conclusion and found that the use of stem tip topping seedlings for cutting is the most effective for the safe overwintering of potato seedlings. One reason may be the differences in the characteristics of growth among varieties. Secondly, there is often a dysregulation of the "source-sink" source and unbalanced growth between the

stems, leaves, and tubers during the process of growth of sweet potatoes. Topping and restricting the length and growth of the main stems is conducive to the transportation of dry matter to the ground, thereby promoting root growth [41,42]. This is consistent with the necessary conditions for the safe overwintering of vegetable sweet potato seedlings, namely, well-developed, sound, and robust roots.

The two factors of density and transplanting period have a certain impact on the overgrowth of leaf-vegetable sweet potatoes. Research by Xu [43] showed that the cutting density in the optimal combination was the highest among the three levels and reached 285,000 plants/ha. In addition, the cutting period also came first among the three batches (24 September 2015). This is similar to the result obtained in this study that showed that the optimal density is the highest among the three levels (250,000 plants/hm$^2$), and the transplanting period also came first (1 November 2021, 21 October 2022). When the overwintering seedlings are cut earlier, this facilitates the utilization of a higher temperature before winter to promote early and rapid root growth, which lays a sound foundation for the safe overwintering of the sweet potato seedlings. Simultaneously, earlier transplantation lessens the impact of low temperature on the selected sweet potato seedlings. The sweet potato seedlings are strong and high quality, which is conducive to safe overwintering. The planting density of sweet potatoes should be determined by the variety, soil, water, and fertilizer conditions, planting period, and planting method [44]. Many Chinese researchers have extensively studied the relationships between cutting density, yield, and quality [45–51]. However, the relationship between planting density and cold tolerance of the overwintering seedlings is currently in the validation stage and merits further study.

IBA can promote cell division and growth and accelerate the formation and growth of the primary roots. Thus, this compound has been widely used to cultivate the seedlings of plants, such as corn, pepper (*Capsicum annuum*), and tobacco (*Nicotiana tabacum*) among others [52–54]. The difference in the mass concentrations of IBA has an impact on the rooting effect, which has been confirmed in flowers, trees, and economic crops [55–57]. Different types of plants respond inconsistently to varying types of rooting agents. Rooting agents have a positive effect at lower concentrations, while a higher concentration can have a negative effect that hinders the normal growth and development of roots. Currently, there have been few studies on the application of IBA in the production of sweet potatoes, particularly leaf-vegetable sweet potatoes. Therefore, it is necessary to strengthen the research in this area in the future.

## 5. Conclusions

The results showed that the use of simple solar greenhouse can successfully implement the safe overwintering of leaf-vegetable sweet potato seedlings in the Changsha area of Hunan province, and provide stable, sufficient and high-quality seed sources for realizing the early market of stems and leaves. The three agronomic traits of stem thickness, root length, and root diameter can be used as key indicators for identifying and evaluating the overwintering growth of leaf-vegetable sweet potato seedlings. Compared with the physiological and biochemical indexes, the method is simple and intuitive, and corresponding cultivation measures can be taken directly for the above three indexes in production, with strong operability. The range analysis showed that A3B3C1D1 (250,000 plants/hm2, stem tip core-picking seedlings, 50 mg/L, the first batch of transplanting) was the best combination for the three key evaluation indexes: stem diameter, root length, and root diameter. After further analysis, it can be concluded that factor D (transplanting time) is the main factor for the two key evaluation indicators of stem diameter and root diameter, while the other three factors have no significant differences. Factor B (selection of cutting seedlings) is the main influencing factor for the key evaluation index of root length. After the analysis of stem variance, it can be concluded that the other three factors have the following impact on root length: transplanting time (D) > rooting agent concentration (C) > planting density (A). In the future, we should base ourselves on the actual situation in Changsha region, guided by advanced sweet potato planting and cultivation concepts. While further

promoting the application of winter cultivation techniques for vegetable sweet potatoes, we should also conduct in-depth research on high-yield cultivation techniques and cold resistance mechanisms in simple greenhouses, and excel in terms of breeding new varieties of cold-resistant vegetable sweet potatoes.

**Author Contributions:** Conceptualization, X.X., X.T. and Z.Y.; writing—original draft preparation, X.X.; Writing—review and editing: Z.Y.; methodology, X.X.; software, X.X. and K.Z.; formal analysis, X.X. and A.Z.; resources, A.Z. All authors have read and agreed to the published version of the manuscript.

**Funding:** This study was supported by the Youth Science Research Project of the Hunan Provincial Department of Education (20B302).

**Data Availability Statement:** The data are included in the article.

**Conflicts of Interest:** The authors declare no conflicts of interest.

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
