# Peer review of "Screening of Indicators to Evaluate the Overwintering Growth of Leaf-Vegetable Sweet Potato Seedlings and Their Main Influential Factors"

_agriculture, doi:10.3390/agriculture14050762_

Round 1
Reviewer 1 Report
Comments and Suggestions for Authors
Dear Authos,
MS can be improved, my comments:
1) The abstract should be improved on the relevance of the study, main results, and suggestions.
2) The introduction can be improved, with more information about previous studies, the relevance of the study, the aim of the study, and references are not new, only 30% of references are new from 2019.
3) In the introduction the authors described about location of the study, this part must be moved to the M%M part.
4) In the M&M part give cite to each method.
5) The results part can be started from the text not the figure.
6) Figure 1 and 2 can be improved.
7) In the 3.4.1. and 3.4.2 part no information about results. It will be added or combined with other subsections.
8) In Table 2, should show the "Treatment type"
9) In the Discussion part a lot of information about results, should be given short information and compared with previous studies. The discussion can be improved by analyzing the same studies of other researchers.
10) The conclusion can be improved by adding main conclusion of the study, recommendation/suggestion for futher studies or plans for future.
11) I add my comments to PDF file.

Author Response
Thank you for taking time out of your busy schedule to review my paper and put forward valuable revision suggestions.
1.1. I have revised the abstract as required.
2.I have revised the introduction as required and added some new documents within 5 years.
3.The introduction of the climate in Changsha has been moved to the Materials and methods section.
4.The title of Figure 1 has been modified and Figure 2 identifies the European distance "2.5".
5.The contents of 3.4.1. And 3.4.2 have been modified.
6.The discussion section has been revised in response to comments.
7.Directions and objectives of future research are added in the conclusion part

Reviewer 2 Report
Comments and Suggestions for Authors
Improve the material and methods, data presentation, discussion, and conclusion.
Check the comments on the manuscript.

English needs to be improved, need to be consistent of using terminology.
Author Response
Thank you for taking time out of your busy schedule to review my paper and put forward valuable revision suggestions.
I made the necessary changes or rewrites to the abstract, introduction, discussion and conclusion of the paper.

Reviewer 3 Report
Comments and Suggestions for Authors
In this manuscript authors address the how to overcome the chilling effects on sweet potato, and suggested that greenhouse condition can be used one of the strategies.
I have few minor concern
1. How you choose the IBA concentration?
2. Figure legends are not clear and there is no explanation.
Comments on the Quality of English LanguageMinor spell check requires
Author Response
Thank you for taking time out of your busy schedule to review my paper and put forward valuable revision suggestions.
I have made some modifications to the English expressions in the paper, especially some professional terms.

Reviewer 4 Report
Comments and Suggestions for Authors
Dear Authors,
The review of the work entitled: "Screening of indicators to evaluate the overwintering growth of vegetable sweet potato seedlings and their main influential factors" indicates the strengths and weaknesses of the work and suggestions for improving the work.
Strengths:
1. Comprehensive analysis: The study was conducted taking into account various aspects such as morphological, physiological and biochemical characteristics, which allowed for a comprehensive assessment of the growth of overwintering sweet potato seedlings.
2. The use of various measurement and statistical analysis methods, such as principal component analysis, membership function, cluster analysis and stepwise regression, allowed for more accurate data evaluation and conclusions.
3. The study identifies optimal conditions, such as greenhouse type, cutting density, cutting location and cutting time, that can be used to safely overwinter sweet potato seedlings.
4. The study results may have practical applications in agricultural and horticultural production, helping farmers adapt their sweet potato cultivation methods for better results.
5. Providing education and information about the study results to farming communities can contribute to better use of new methods and practices in sweet potato cultivation.
Weaknesses:
1. Repetition: Although the study was performed with three repetitions, repetition may be limited, which may affect the reliability of the results.
2. Local limitations: The study was conducted in a specific location (Changsha, Hunan Province, China), which may limit the generalizability of the results to other regions.
3. Lack of comparison with other varieties: The study focused on one sweet potato variety (Fucaishu 18), which limits the ability to compare the results with other varieties.
4. Lack of analysis of the impact of environmental conditions: The study mainly focuses on factors manipulated by researchers (such as cutting density, cutting location) rather than on natural environmental conditions that may also affect sweet potato growth.
5. Lack of long-term evaluation: The study is limited to the period from October to February, which may not take into account the long-term effects of the sweet potato cultivation methods used.
Suggestions for improvement
1. Edit the abstract in accordance with the requirements of the MDPI publishing house and the Agriculture magazine.
2. Extending the analysis of factors influencing the tested features: The study may take into account a wider range of factors, such as soil composition, applied fertilizers, lighting, irrigation, which may have a significant impact on the growth and yield of sweet potatoes.
3. Taking into account economic aspects: The study can be extended to assess the economic efficiency of the sweet potato cultivation methods used, which is important for sweet potato producers.
4. Conclusions should contain at least one point looking towards the future.
Author Response
Thank you for taking time out of your busy schedule to review my paper and put forward valuable revision suggestions.
1. I have revised the abstract of my paper.
2. Before the experiment, I tested the soil fertility and designed the fertilization program based on it. In the next step, I plan to carry out wintering tests on fertilization, light, yield, quality and other aspects based on this experiment.
3. In the introduction, I have added some explanations about the economic benefits of vegetable sweet potatoes. At present, vegetable sweet potato has broad market prospects and good economic benefits. In Changsha, the market sales price can reach 16 yuan per kilogram in February-March every year. This is also the reason why I want to carry out the winter test, that is, to realize the annual production of Leaf-vegetable sweet potato, and try to put it on the market in advance.

Reviewer 5 Report
Comments and Suggestions for Authors
Congratulations for your study. Τhis particular research is very interesting. However, I believe that you need to revise some elements of your research, especially the presentation of the results. All notes included in the file.

Author Response
Thank you for taking time out of your busy schedule to review my paper and put forward valuable revision suggestions.
1. I have modified the keywords according to your suggestions
2. References are added to the places that need to be explained, such as the climatic characteristics of Changsha
3. Tables 5 and 6 of your submission have been reversed, explaining 2021-2022 first and then 2022-2023
4. Part 4.4.2 has been revised to introduce and add other scholars' views for discussion and confirmation
